# Helminth Community Structure of Tits *Cyanistes caeruleus* and *Parus major* (Paridae) during Their Autumn Migration on the Southern Baltic Coast

**DOI:** 10.3390/ani13030421

**Published:** 2023-01-26

**Authors:** Izabella Rząd, Anna Okulewicz, Rusłan Sałamatin, Magdalena Szenejko, Remigiusz Panicz, Jarosław K. Nowakowski, Agata Stapf

**Affiliations:** 1Institute of Marine and Environmental Sciences, University of Szczecin, Wąska 13, 71-415 Szczecin, Poland; 2Molecular Biology and Biotechnology Centre, University of Szczecin, Wąska 13, 71-415 Szczecin, Poland; 3Department of Parasitology, Faculty of Biological Sciences, University of Wrocław, Przybyszewskiego 63, 51-148 Wrocław, Poland; 4Department of General Biology and Parasitology, Medical University of Warsaw, Chałubińskiego 5, 02-004 Warsaw, Poland; 5Department of Microbiology and Parasitology, Faculty of Medicine, Collegium Medicum, Cardinal Stefan Wyszyński University in Warsaw, Kazimierza Wóycickiego 1/3, 01-938 Warsaw, Poland; 6Department of Meat Science, Faculty of Food Science and Fisheries, West Pomeranian University of Technology, 4 Kazimierza Królewicza Street, 71-550 Szczecin, Poland; 7Bird Migration Research Station, University of Gdańsk, Wita Stwosza 59, 80-308 Gdańsk, Poland; 8Department of Biological Sciences, Faculty of Sport Science in Gorzów Wielkopolski, Poznan University of Physical Education, Estkowskiego 13, 66-400 Gorzów Wielkopolski, Poland

**Keywords:** Baltic Sea, bird migration, filarial nematodes, *Cardiofilaria*, *Cyanistes*, *Diplotriaena*, helminths, *Leucochloridium*, *Parus*

## Abstract

**Simple Summary:**

Parasites of birds can be an indicator of environmental quality, but this issue is very poorly understood. Migratory birds can transfer parasites between ecosystems, thus increasing the size of their populations. The biodiversity of bird parasites and the predominance of specific parasite species can be influenced, as in the case of non-parasitic animals, by global changes in the environment, including climate change and anthropogenic transformation of ecosystems. The aim of the present research was to describe and compare helminth communities of Eurasian blue tit *Cyanistes caeruleus* and the Great tit *Parus major*, at the component community and infracommunity level, on the southern coast of the Baltic Sea during autumn migration to their wintering grounds. The main findings of the research were the presence of several parasite species that had not previously been recorded in these tits in Poland, the dominance of the digenetic fluke, and the high prevalence of the filarial nematodes.

**Abstract:**

The research problem undertaken in this study is to determine the scale of infection of Eurasian blue tit *Cyanistes caeruleus* and Great tit *Parus major* and the biological diversity of their internal parasites, helminths. The aim of the study is to gain new knowledge about the structure of the helminth communities of the Eurasian blue tit and Great tit on the southern coast of the Baltic Sea during autumn migration to their wintering grounds. Helminths of tits were collected in 2008–2012 on the southern coast of the Baltic Sea in Poland. PAST v. 2.11 software was used for the calculations. Barcoding DNA was used to identify trematodes initially classified based on morphological characters to the genera *Leucochloridium* and *Urogonimus.* Cestodes *Anonchotaenia globata* were recorded for the first time in Poland. The Eurasian blue tit is a new host in Poland for three species of helminths: cestode *Monosertum parinum* and filarial nematodes, *Cardiofilaria pavlovskyi,* and *Diplotriaena henryi*. The Great tit is a new host in Poland for trematode *Urogonimus macrostomus*, cestode *A. globata* and *M. parinum,* and filarial nematode *Diplotriaena obtusa.* The nematode *C. pavlovskyi* was the species most frequently recorded in both host species. A high degree of similarity was found between the component communities and infracommunities of helminths in Eurasian blue tit and Great tit. The new information provided in this study has increased our knowledge of the transmission of helminths in Central Europe.

## 1. Introduction

Birds are hosts for a variety of internal parasites, which can cause disease, poor body condition, and death. Wild migratory birds transmit parasites between natural, semi-natural, and artificial ecosystems, including agricultural habitats. Global changes, especially climate change that causes the transformation of biocoenoses and ecosystems, necessitates scientific research to increase our knowledge of bird migration [1,2]. The process of parasite transmission between ecosystems impacted by global environmental changes, including those caused by pollution, and the associated possibilities or lack of possibility for the further development of populations of various species of parasites are very poorly understood. The increased risk of malaria in birds due to climate change has been documented in recent years [3,4,5,6], but the spread of helminths in the populations of many species of migratory Passeriformes is little known [7]. The need for research on migratory birds as a reservoir of parasites, circulator of parasites in various areas, and the potential risk of parasites and vector-borne diseases has been emphasized by authors such as Fuller et al. and Gruinau et al. [2,8]. Climate change is increasing the importance of vector-borne diseases and pathogens in both humans and animals, e.g., the West Nile virus, protozoa causing malaria, filarial nematodes *Wuchereria bancrofti*, and *Setaria* spp. The migration of tits and their parasites does not translate directly to human parasites. However, mechanisms of parasite transmission by vectors, influenced by global environmental changes, are common and related to the concept of “One Health”, i.e., healthy animals–healthy environment–healthy humanity; thus, those pathogens have to be monitored.

Parasites of Eurasian blue tit *Cyanistes caeruleus* (Linnaeus, 1758) and Great tit *Parus major* Linnaeus, 1758 (Passeriformes, Paridae) are partially known, but the community structure of the helminths occurring in these host species during their autumn migration along the southern Baltic Sea coast is still unknown. This migration proceeds from the breeding grounds to the wintering grounds. Eurasian blue tit and Great tit are common birds throughout Europe. In Poland, both species have the status of numerous breeding species, and the Great tit is the fourth most numerous breeding species [9,10]. Northern and northeastern European populations, including those occurring in Poland, are partially migratory. Great tit and Eurasian blue tit, which migrate in autumn over the southern Baltic coast in Poland, have their breeding grounds in northeastern Poland, Kaliningrad Oblast (Russia), western and central Lithuania, Latvia, Estonia, and the Leningrad Oblast. The wintering grounds of both tit species include northwestern Poland, northern Germany, the Netherlands, and Belgium, and in the case of Great tit, also northeastern France. Autumn migration over the southern Baltic coast begins in mid-September and lasts about 40 days [11]. Ornithological research involving the ringing of these birds is carried out at several research stations in Poland, e.g., by Operation Baltic (https://operbalt.ug.edu.pl/ accessed on 1 August 2020). Analysis of re-trapped birds, i.e., birds recaptured in nets (for a second time or multiple times), at the Bukowo-Kopań bird ringing station (16°25′ E 54°28′ N) has shown that the Great tit and Eurasian blue tit spend from one day to about a week in this area, located on the autumn migration route, or in extreme cases from 0 days to about two weeks (*P. Busse*—personal communication based on Operation Baltic research, 7 August 2021), and the average migration speed is about 30 km/day [12].

The aim of the present research is to describe and compare helminth communities of the Great tit and Eurasian blue tit, at the component community and infracommunity level, on the southern coast of the Baltic Sea during their autumn migration to their wintering grounds. The helminth community structure is assumed to be similar in both tit species due to their close phylogenetic relationship, large population size, similar way of life, and similar food preferences, which can include the intermediate hosts of parasites. However, there are reasons to believe that the structure of helminths may differ in the two species; for example, they prefer different biotope types and feed in different ways. The biotopes inhabited by the Great tit include poor environments, such as coniferous forests and urban areas, while the Eurasian blue tit chooses biotopes more closely associated with deciduous forests and parks. The Great tit readily feeds on the ground, while the Eurasian blue tit avoids feeding on the ground, preferring to penetrate tree crowns and shrubs.

## 2. Material and Methods

### 2.1. Sampling

The birds were collected during the autumn migration season, from September to November 2008–2012. A total of 67 Eurasian blue tit individuals and 166 Great tit individuals were subjected to parasitological analysis. Age was established in 30 Eurasian blue tits (16 juveniles and 14 adults) and 95 Great tits (73 juveniles and 22 adults). Sex was determined in 30 Eurasian blue tits (17 females and 13 males) and 101 Great tits (58 females and 43 males), although in case of some individuals, both age and sex, or only one of the features, were determined. The birds inhabited northern Poland and were collected along the southern coast of the Baltic Sea. Most birds were obtained during sampling organised by Operation Baltic. No birds were sacrificed for the purpose of sampling. The migration process may deteriorate body condition and increase the mortality rate of birds due to numerous and carious factors [13,14,15]. Birds used for the research had died of various causes, e.g., due to a sudden and extremely unfavourable change in weather conditions on the Baltic coast, attack by predators, or collision with an obstacle, such as a window. The birds for parasitological examination were not classified according to the cause of death, and their body condition was not evaluated. Dead birds were not always fresh, as all available birds were used for testing in accordance with the literature recommendations [14,16,17,18], but the internal organs of all birds had adequate conditions for collecting parasites. The freshness of the birds was primarily a factor for cestodes, which are delicate and highly susceptible to lysis, while decomposition had a relatively minor effect on trematodes and nematodes, specimens of which were in very good condition for species identification. For this reason, not all cestode specimens were precisely identified by genus and species. Birds in which cestode species could not be identified or for which other data were unavailable were excluded from the analysis of ecological indices describing the structure of the component community and infracommunity of helminths. All procedures were in compliance with Polish law. Consent for the collection of dead birds and parasitological research was obtained from the General Director for Environmental Protection and the Regional Director for Environmental Protection in Szczecin.

### 2.2. Sample Processing and Parasite Identification

Parasites were preserved in 70–75% ethanol. Trematodes were stained with alum carmine and cleared in clove oil, and microscope slides were mounted with Canada balsam. Cestodes were stained with acetocarmine. Scolices were cleared in Faure’s medium. Nematodes were cleared in glycerol. Morphological examination and identification of parasites were carried out using keys and original works [19,20,21,22,23,24,25,26]. DNA barcoding was used to identify 31 trematode specimens initially classified on the basis of morphological characters to the genera *Leucochloridium* and *Urogonimus* (Leucochloridiidae); further, identification was not possible due to the high similarity or variability of the main diagnostic features within the genera (a wide range of body shape variation, the size ratio of suckers, overlapping of internal organs, or their concealment behind uterine loops filled with dark eggs in adult trematodes). For barcoding, genomic DNA was isolated from each individual using the High Pure PCR Template Preparation Kit (Roche Diagnostics GmbH, Mannheim, Germany). Next, a fragment (~300 bp) of cytochrome c oxidase subunit I (COI) was amplified using primers JB3 and JB4.5 [27] and JB3 and JB4.5-in [28]. The composition of 20 μL of the reaction mixture was as follows: 5x Phire Reaction Buffer with Mg^2+^, 0.2 mM dNTP, 0.5 µM of each primer, 0.4 µL Phire Hot Start II DNA Polymerase (Thermo Fisher Scientific, Waltham, MA, USA), and 6.0 μL template DNA. PCR reactions were carried out in a T100TM Thermal Cycler (Bio-Rad, Hercules, California, USA) in 40 cycles, according to the following temperature profile: initial denaturation at 98 °C for 5 min, then 98 °C for 5 s, 50 °C for 5 s, 72 °C for 20 s, with final elongation at 72 °C for 1 min. Individual PCR products were identified on a 1.5% agarose gel in the presence of the Nova 100 bp DNA Ladder mass standard (Novazym, Poznań, Poland). DNA sequencing of all PCR products was performed by Genomed, Poland, and the raw reads were assembled with Geneious 8.0 [29] and compared against the GenBank sequences using BLAST [30]. Species identity of the flukes from this study was confirmed based on the alignments with COI homologous sequences retrieved from the GenBank database using BLAST search [30]. Then, the genetic diversity of the flukes (number of haplotypes, haplotype diversity, and nucleotide diversity) was calculated using DNAsp 5.10 software [31]. The ML tree was constructed using MEGA5 based on the sequences of haplotypes of *Parus major* and *Cyanistes caeruleus* flukes identified in this paper and sequences of other flukes retrieved from the GenBank [32]. The *Ornithobilharzia canaliculata* (GenBank acc. no. AY157194.1) COI sequence was used as the outgroup. 

### 2.3. Data Analysis

Basic parasitological parameters were calculated: prevalence, as the percentage of infected hosts among all tested birds; mean intensity, i.e., the number of parasites divided by the number of infected hosts; range of intensity, i.e., the minimum and maximum number of parasites; and mean abundance, i.e., the number of parasites divided by the number of all hosts, infected and not infected. All parameters were defined according to Bush et al. [33]. The component community structure was analysed, specifying the community of all parasites in the Eurasian blue tit population and all parasites in the Great tit population, as well as the infracommunity structure, meaning the community of parasites in one host individual. For each component community, biological diversity was determined using Simpson’s index and dominance using the Berger–Parker index. The Brillouin index was used for determination of infracommunity diversity. All indices are defined according to Magurran [34]. PAST v. 2.11 software [35] was used for the calculations. Parasitological parameters and ecological indices of the Eurasian blue tit and Great tit were compared. Parasitological parameters were also compared by sex and age of birds. Prevalence was compared using the chi^2^ test. The nonparametric Mann–Whitney U test was used to test differences in the intensity, abundance, richness, and Brillouin index between communities of the Eurasian blue tit and Great tit. Statistical tests were carried out in Statistica 13.1 software.

## 3. Results 

### 3.1. Faunistic and Molecular Analyses

New faunistic data were obtained in this research. Cestodes *Anonchotaenia globata* (von Linstow, 1879) were recorded for the first time in Poland. The Eurasian blue tit is a new host in Poland for three helminth species: *Monosertum parinum* (Dujardin, 1845), *Cardiofilaria pavlovskyi* (Strom, 1937), and *Diplotriaena henryi* (Blanc, 1919.) The Great tit is a new host in Poland for *Urogonimus macrostomus* (Rudolphi, 1802), *A. globata*, *M. parinum,* and *Diplotriaena obtusa* (Sonin, 1968) (Table 1).

The molecular analyses conclusively identified the trematodes with atypical morphological characters to two species: *Leucochloridium paradoxum* (Carus, 1835) (n = 21) and *U. macrostomus* (n = 10) (Figure 1). Their taxonomic identity was confirmed in GenBank at the level of identity of alignments to homologous sequences above 98%. Maximum likelihood estimation also showed that in *P. major* individuals (nos. 4 and 7), there was only one species of fluke (letter designations a, b, etc.). The assessment of genetic diversity showed a greater number of haplotypes (n = 6) and lower levels of haplotype diversity (0.48 ± 0.13) and nucleotide diversity (0.0026) in trematodes *L. paradoxum* than in specimens of *U. macrostomus* (n = 4, 0.64 ± 0.15 and 0.0144, respectively). Further analysis showed that all haplotypes of *U. macrostomus* (PL-US-HU1–PL-US-HU4) and five of the six haplotypes of *L. paradoxum* (PL-US-HL1–PL-US-HL5) had not previously been described, so they were deposited in the GenBank database (MK882500-MK882508). 

### 3.2. Structure of the Component Communities and Infracommunities of Helminths

Among parasite species, only the prevalence of *L. paradoxum* showed statistically significant differences between component communities of the tit species (*p* = 0.027) (Table 1). The prevalence of helminths was higher in the Great tit than in the Eurasian blue tit (62.7% vs. 44.8%, *p* = 0.012) (Table 2).

The mean intensity and mean abundance of helminths were higher in the Great tit than in the Eurasian blue tit, but only the difference in mean abundance was statistically significant (11.4 vs. 6.0, *p* = 0.024). There were five helminth taxa parasitizing the Eurasian blue tit and eight in the Great tit (Table 2). The component community and infracommunity of helminths of the Eurasian blue tit and Great tit did not differ; Simpson’s index, the Berger–Parker index, and Brillouin index were similar in the two species of birds and indicated strong dominance of *L. paradoxum* (Table 2).

### 3.3. Helminth Infection According to Host Age and Sex

In the juveniles of Eurasian blue tit, trematodes and nematodes were identified, while in the adult birds, additional parasitic cestodes were found. The differences between parasitological parameters in juvenile and adult Eurasian blue tits were not statistically significant (Table 3). 

Trematodes, cestodes, and nematodes were found in juvenile and adult Great tits, but the prevalence of cestodes was much higher in adult Great tits (31.8%) than in juveniles (2.7%), and the difference was statistically significant (*p* = 0.000) (Table 3). Trematodes, nematodes, and cestodes were found in female Eurasian blue tits, but males were infected only by nematodes. The differences between parameters of infection of male and female Eurasian blue tits were not statistically significant. Trematodes, cestodes, and nematodes were present in both female and male Great tits, and the differences between parameters of infection in females and males of Great tits were not statistically significant (Table 3).

## 4. Discussion

The presence of parasites as factors regulating the size of host populations can weaken the host organism. Clinical symptoms and the mortality of birds caused by parasites are noted in the literature [36,37]. Although it is widely believed that in the natural environment, parasites do not cause severe pathological changes in wild animals, in contrast to livestock and humans, it seems unlikely that several hundred or even a few dozen trematodes of the genus *Leucochloridium* in a bird’s gut, *Cardiofolaria* nematodes in the pericardium and coelom, or *Diplotriaena* in the pulmonary cavity would not disturb the function of these organs or affect the condition of seasonally migrating birds. For example, large numbers of flukes *L. paradoxum* in the cloaca of birds have been reported in the Czech Republic, with the suggestion that this may lead to emaciation and the consequent death of the birds [36].

### 4.1. Similarities and Differences in Infection of Eurasian Blue Tits and Great Tits: Different Parasite Composition but Similar Helminth Community Structure

The present results showed that during the migration of Eurasian blue tit and Great tit over the southern Polish coast of the Baltic Sea, the helminths of the two species have a different species composition but similar community structure. The study revealed seven parasite–host relationships previously unknown in Poland: between Eurasian blue tit and nematodes (*C. pavlovskyi* and *D. henryi*) and cestodes (*M. parinum*), and between Great tit and nematodes (*D. obtusa*) trematodes (*U. macrostomus*) and cestodes (*M. parinum* and *A. globata*). These parasites had previously been recorded in Poland in other bird species [38]. Among the nine identified helminth species, four were common to both tit species: *L. paradoxum*, *M. parinum*, *C. pavlovskyi,* and *D. henryi*. The dominant species in the communities of the tits in terms of prevalence was *C. pavlovskyi*, while *L. paradoxum*, whose prevalence can be described as intermediate, was clearly dominant in terms of the abundance of individuals in the helminth communities. The values for the parasitological parameters of *M. parinum* and *D. henryi* were low, like those of other helminths found only in the Eurasian blue tit (*Plagiorchis maculosus* (Rudolphi, 1802)) or only in the Great tit (*U. macrostomus*, *A. globata*, *Capillaria triedens* (Dujardin, 1845) and *D. obtusa*), which should be regarded as accessory species. 

The presence of *L. paradoxum* and *U. macrostomus* among trematode specimens that could not be assigned to species on the basis of morphological characters was confirmed by molecular analysis. Trematodes *L. paradoxum* and *U. macrostomus* are typical species found in tits in Poland and other European countries. The DNA barcoding method used for genetic identification, based on analysis of variation in the *COI* gene sequence, proved successful in resolving doubts regarding the identification of trematodes found in Eurasian blue tit and Great tit as *L. actitis* and *L. macrostomum*. The results of the molecular analysis clearly confirmed the presence of two species of trematodes: *L. paradoxum* and *U. macrostomus*. *L. paradoxum* was shown to have more haplotypes and lower haplotype and nucleotide diversity than *U. macrostomus*. The differences were most likely due to the fact that the number (n = 21) of *L. paradoxum* trematodes identified was higher than the number of trematodes of the species *U. macrostomus* (n = 10). The genetic analyses included in this study were mainly meant to identify the species of trematodes that could not be classified on the basis of morphological characteristics. Therefore, confirmation and verification of the genetic diversity among the collected representatives of *L. paradoxum* and *U. macrostomus* require more extensive analyses based on a larger group of representatives of both species.

Following the views of Freeland [39] regarding factors determining the distribution of parasitic species in potential host species, i.e., phylogenetic origin, size, morphology, and nutritional requirements, several possible reasons for the similarities and differences in the infection of the Eurasian blue tit and Great tit can be postulated. The similarities, i.e., the high prevalence of *C. pavlovskyi*, the lack of significant difference in the intensity of helminth infection in the two tit species, the low Simpson’s index, the high Berger–Parker index indicating the dominance of *L. paradoxum* in the communities of the two host species, the lack of significant difference between the diversity of the helminth infracommunity of the Eurasian blue tit and Great tit, and the lack of significant difference between the number of helminth species in the helminth infracommunity of the two tit species, may have several explanations:--Both species of birds prefer similar biotopes, such as forests, parks, and gardens, and they can both be found in the same areas.--The birds’ food and feeding behaviour are very similar, and their nesting and breeding strategy are essentially identical.--At their wintering grounds, the two species form mixed flocks and winter in exactly the same places.--The origins of the captured populations and their flight routes are the same, except that, in general, that of Eurasian blue tits is statistically shorter.--Nematodes *C. pavlovskyi* and trematodes *L. paradoxum* have broad host specificity and are known as parasites of many birds of the order Passeriformes.

Filarial nematode *C. pavlovskyi* is a generalist parasite, found mainly in passerines but also in other birds [21,40,41,42]. Its life cycle is indirect and includes a dipteran of the genus *Armigera*, *Culex*, or *Mansonia* [43]. Scientists currently consider the changing ranges of mosquitoes (Culicidae) as one of the effects of climate change. This interrelation is explored more extensively in the study of mosquito-borne human diseases and falls within the scope of medical parasitology. Another important parasite in terms of prevalence and intensity of infection is the digenetic trematode *L. paradoxum*. Birds are infected by these flukes by pecking the eye stalks of infected snails *Succinea putris* [44,45] that contain sporocysts with metacercariae of these parasites, whose appearance lures feeding birds. 

The differences between the helminth community structure of the Great tit and Eurasian blue tit, such as the larger number of helminth species in the Great tit, significantly different prevalence of helminths between the two species, significantly different abundance of helminths, and significantly different prevalence of *L. paradoxum* between the two host species, can be explained by the following factors:-The larger number of helminth species found in the Great tit was linked to the larger sample size; more than twice as many Great tits were examined as Eurasian blue tits.-Although both species prefer similar biotopes, the Great tit shows greater ecological tolerance regarding habitat conditions, which means that it may also inhabit poorer environments, e.g., coniferous forests or urban areas. The Eurasian blue tit is more closely associated with broadleaf forests and parks. A key to understanding the differences may be their feeding sites. The Great tit prefers feeding on the ground, while the Eurasian blue tit avoids foraging on the ground and prefers to penetrate tree crowns and shrubs.-The Great tit’s higher tolerance for habitat conditions may result in more frequent contact with parasites present in the environment, as well as with their intermediate hosts, which may partly explain the significantly higher prevalence of helminths, mainly *L. paradoxum* (but also *U. macrostomus*), and the significantly higher mean abundance of helminths in the Great tit population than in the Eurasian blue tit population.

### 4.2. Presence of Parasites in the Geographical Distribution of C. careuleus and P. major

Our study characterised the diversity and abundance of internal parasites in the tits during the period of their autumn migration. However, the presence of the parasites can not be explained solely by the fact of bird migration since parasites may be present both at the nesting and wintering sites that largely overlap. The presence of parasites in tits does not stem from the migration itself but from their distribution in the habitat area of tits. However, due to the commonly observed clumped density of parasites in hosts and ecosystems, birds migrating between those areas can be invaded by diverse species and a number of pathogens. Therefore, birds that do not migrate and inhabit overlapping nesting and wintering areas can have different abundances and species diversity of parasites than birds that migrate. At this time point, the comparison of parasite populations for *C. caeruleus* and *P. major* is hindered due to a lack of detailed information on the parasites of tits during nesting and wintering periods. Among the parasites identified in our study, the high incidence of *C. pavlovskyi* is presumably related to the migration of birds. In 1971–1988, this species of nematode was identified in *P. major* nesting in Poland with a prevalence of 6% and a range of intensity 1–15 [46]. Regarding the number of parasites in hosts, expressed by mean intensity or mean abundance, the high values might result from environmental and physiological stress, but not only from the superinfective processes [47].

Research to date on the Eurasian continent indicates that within its range, the Great tit may host about twice as many helminth species as the Eurasian blue tit (Table 4) [25,26,36,48,49,50,51,52,53,54,55,56,57,58,59,60].

The most common have been species of trematodes, followed by cestodes and nematodes. Data from outside Poland are primarily from the Czech Republic, Eastern Europe, the Asian part of Russia, Ukraine, Finland, and Slovakia [25,26,36,48,49,50,51,52,53,54,55,56,57,58,59,60]. Bird migration plays a major role in the transmission of parasites between ecosystems, but this process is little known in songbirds, in contrast to wetland and water birds [61,62]. The range of most bird helminth species identified in this study is Palaearctic, and some of them are cosmopolitan. Migratory birds often are infected by new species of parasites in ecosystems located along the migration route and at the wintering grounds and transfer them to their breeding grounds, as well as from the breeding grounds to the wintering grounds. In autumn, many species of birds fly along the Baltic coast from their breeding grounds toward their wintering grounds. The association between avian blood parasites and the migration of birds is well known [4], but the association between helminths and these migratory birds is not. In autumn of the years 2008–2012, a total of 99,637 birds, including 38,917 Great tits and 7320 Eurasian blue tits, were captured and ringed at Operation Baltic stations on the southern coast of the Baltic Sea in Poland. Parasitological parameters determined in a research sample of these birds indirectly provide information on the occurrence of parasites in the entire migratory populations of the Eurasian blue tit and Great tit. The geographic distribution of parasites essentially overlaps with the range of their hosts (see: faunaeuropea.eu [63]). Parasitological parameters reflect the parasitic fauna of Great tits and Eurasian blue tits immediately before their migration, but these parameters can also be influenced by parasites acquired at stopover sites after the migration has begun. The Eurasian blue tit and Great tit migrate across short distances within Europe. The various parasite species found in the present study had previously been recorded in tits or other bird species in Eastern Europe (e.g., northeastern Poland) and in the Asian part of Russia, which may overlap with the breeding grounds of tits. They have also previously been recorded in Western and Southern Europe (e.g., northwestern Poland, the Czech Republic, Slovakia, Romania, Bulgaria, France, and Spain) [46], where seasonal migration of tits takes place or which serve as wintering grounds for these birds. The parasitological parameters of the tit parasites detected in the present study indicate that autumn migration is conducive mainly to the development of the populations of trematodes *L. paradoxum* and nematodes *C. pavlovskyi*, while other parasite species are rare in tits.

The tapeworm *A. globata* (Paruteniridae, Cyclophyllidea) was recorded in the present study for the first time in Poland. *A. globata* is widespread in many countries in Europe, including those along the migration route of tits, e.g., Germany, France, and Moldova [63]. Poland had previously been a gap on the map of Europe, which was filled owing to the research undertaken in this study. *M. parinum* is a cestode that is commonly recorded. It has previously been noted in Poland in *Sturnus vulgaris* [38]. The trematode *L. paradoxum* is a parasite of passerines and waders in Europe [64]. In Poland, this trematode has previously been recorded in *Turdus merula*, *T. philomelos*, *C. caeruleus,* and *P. major*, as well as in intermediate hosts, in practically all of Poland, including the migration route of tits from northeastern to northwestern Poland, together with the Baltic coast [38,49,50]. Outside of Poland, it has been recorded in Germany, Great Britain, Denmark, Finland, Moldova, Ukraine, Norway, Sweden, and the Czech Republic [45,48,63]. As indicated in the abovementioned works, trematodes of *L. paradoxum* are commonly present in Europe, including Poland. Thus, the identifications of *L. paradoxum* in migrating tits, both in the earlier studies and in our work, do not indicate that the migration of birds is the factor that influences infection of *L. paradoxum* but only shows that parasites are present in the birds during their migration.

*U. macrostomus* is a parasite of passerines in the Holarctic region. In Poland, it is noted in numerous bird species on the Baltic Sea coast as well as in the south of the country [38]. Outside of Poland, it has been recorded in many European countries, e.g., Moldova, Germany, France, Norway, and the Czech Republic [48,63]. *P. maculosus* is a trematode with broad host specificity in the Holarctic region. It is a frequent parasite of birds of several orders, including Passeriformes. It has also been noted in India, the Philippines, and Australia (according to Sitko et al. [48]). In Poland, it has been recorded in several species of birds on the Baltic Sea coast and in other parts of the country [38]. In Europe, it has been found in Lithuania, Moldova, Germany, and France [63]. The nematode *C. pavlovskyi* has Holarctic distribution and is known from Europe, Asia, and North America [46]. In Poland, it has previously been found in several *Turdus* species and in *Parus major* on the Baltic coast, as well as in southern Poland [40,46]. Outside of Poland, its presence has been noted in the northeast in the Kaliningrad Region, Latvia, France, and Spain [63]. *C. pavlovskyi* has also been recorded in birds in Ukraine [42]. The presence of *C. triedens* was previously reported in Europe among Passeriformes and birds of other orders, as well as in North America, Central America, and Asia. This ringworm had been recorded in Poland in several species of birds, including the Great tit, and outside of Poland in France and Spain [26]. The presence of *D. henryi* was previously recorded in Europe in Passeriformes, including Great tit, and in birds of other orders—in Poland, but also in Asia and Africa. In Europe, it has been recorded in birds of various orders in France, Romania, and Bulgaria [46]. The nematode *D. obtusa* has been reported in northeastern and central Poland [40]. In Europe, the presence of *D. obtusa* has been noted mainly in the northeast (Kaliningrad, Moldova) and south (Bulgaria, Italy, Ukraine), but also in the north (Netherlands) [63].

### 4.3. Similarities and Differences in Juvenile and Adult Tits and in Males and Females

The number of parasite species in adult Eurasian blue tits is greater than in juveniles, while in the Great tit, it is the same in adults and juveniles. In adult birds, an infection can take place at both the breeding grounds and the wintering grounds, which for both species cover northwestern Poland, northern Germany, the Netherlands, and Belgium, and in the case of Great tit, also northeastern France. Therefore, the parasite fauna of adult birds may be partially mixed. Helminths occurring in juvenile birds migrating in autumn for the first time will mainly represent local parasites present in the host breeding grounds, but it cannot be ruled out that infection with these parasites could take place during autumn migration. Most of the differences between the parameters of infection of juvenile and adult tits were not statistically significant, with the exception of the prevalence of cestodes, which was significantly higher in adult Great tits than in juveniles. Differences in infection depending on the age of the host are well documented in the literature for various animal species e.g., [62]. The higher prevalence of tapeworms in adult Great tits is linked to the more frequent share of the intermediate hosts of these parasites in their diet than in juvenile birds.

The number of parasite species was higher in females of both tit species than in males (Table 3), but there were no significant differences between the parameters of infection of females and males in the two species of tit. The results of studies by various authors on the relationship between infection and the sex of the host are inconclusive. The relationship between parasite infection and host sex may depend on other environmental factors [65,66]. Experimental research by Richner et al. [65] showed that in males from Great tit families in which the number of nestlings in the nests was experimentally increased, the rate of infection with malaria parasites was twice that of males from families in which the number of nestlings was not increased. This was explained by the fact that the chicks were fed by the males, not the females [65]. Studies of Blackbird *Turdus merula* have shown the influence of an anthropogenic environment on differences in helminth infection of males and females. Differences between the infection parameters of males and females were observed in an urban Blackbird population but not in a forest population [62].

## 5. Conclusions

The new information provided in this study has increased our knowledge of the transfer of helminths during the migration of birds in central Europe, and the Eurasian blue tit and the Great tit were shown to be important carriers of filarial nematodes *Cardiofilaria pavlovskyi, Diplotriaena henryi,* and *Diplotriaena obtusa*. This research is necessitated by gaps in current knowledge of changes taking place in the population of helminth species in the context of global environmental changes, especially climate change. Similarities and differences in helminth communities of the Great tit and Eurasian blue tit were demonstrated, with a greater spread of *Leucochloridium paradoxum* observed in the population of the Great tit than in that of the Eurasian blue tit. There is a need to monitor the size of communities and populations of helminth species such as *L. paradoxum*, whose high intensity of infection may affect the health and body condition of birds. Based on the results of the study, we recommend increased research on the occurrence of parasitic helminths in migratory birds, examination of birds for filarial nematodes, and studies of populations of different species of *Leucochloridium* in the environment.

## Figures and Tables

**Figure 1 animals-13-00421-f001:**
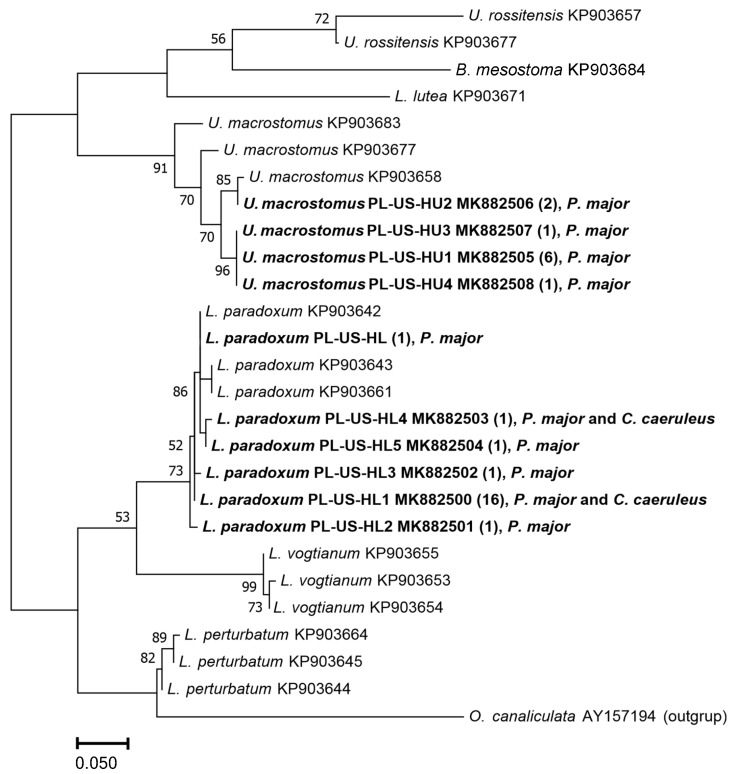
Maximum likelihood analysis of sequences of the COI loci in the mitochondrial DNA of flukes *Parus major* and *Cyanistes caeruleus.* Bootstrap values expressed as percentages from 1000 replicates are given next to each branch. Values less than 50% are not shown. Results (haplotypes, hosts) obtained in this study are in bold. Numbers of flukes with a given haplotype are given in brackets.

**Table 1 animals-13-00421-t001:** Parasitological indicators of helminths in Eurasian blue tit and Great tit during autumn migration on the southern coast of the Baltic Sea, northern Poland, 2008–2012. Numbers of examined birds are given in brackets. Total values relate to the parasites identified up to the species level and individuals with no taxonomic assignment (details in the Materials and Methods).

Helminth	Eurasian Blue Tit *Cyanistes caeruleus* (67)				Great Tit *Parus major* (166)			
	Prevalence [%]/n	Mean Intensity	Range of Intensity	Mean Abundance	Prevalence [%]	Mean Intensity	Range of Intensity	Relative Abundance
*Leucochloridium paradoxum*	6.0/4 *	81.7	5–154	4.9	16.9/28 *	56.5	2–220	9.5
*Urogonimus macrostomus*	0	-	-	-	2.4/4 ^NH^	18	1–60	0.4
*Plagiorchis maculosus*	1.5/1	1	-	0.0	0/0	-	-	-
Digenea—total:	7.5/5 *	65.6	1–154	4.9	19.3/32 *	51.7	1–220	10.0
*Anonchotaenia globata*	0	-	-	-	0.6/1 ^NGR, NH^	2	-	0.0
*Monosertum parinum*	1.5/1 ^NH^	10	-	0.1	0.6/1 ^NH^	5	-	0.0
Cestoda—total:	6.0/4	3.5	1–10	0.2	11.4/19	3.7	1–10	0.1
*Cardiofilaria pavlovskyi*	32.8/22 ^NH^	2.1	1–6	0.7	44.0/73	2.6	1–9	1.1
*Capillaria tridens*	0	-	-	-	0.6/1	1	-	0.0
*Diplotriaena henryi*	1.5/1 ^NH^	2	-	0.0	0.6/1	6	-	0.0
*Diplotriaena obtusa*	0	-	-	-	0.6/1 ^NH^	1	-	0.0
Nematoda—total:	37.3/25	2.3	1–6	0.8	45.8/76	2.7	1–14	1.2

n, number of infected birds; NH, new host in Poland, NGR, new geographical record, *, statistically significant difference between parameter of infection of Eurasian blue tit and Great tit, *p* ≤ 0.05.

**Table 2 animals-13-00421-t002:** Structure of helminth communities of Eurasian blue tit and Great tit during autumn migration on the southern coast of the Baltic Sea, northern Poland, 2008–2012. Numbers of examined birds are given in brackets.

	Component Community								Infracommunity	
Host	Prevalence [%]/n	Mean Intensity of Infection	Range of Intensity	Mean Abundance	Number of Parasite Species	Simpson’s Index (1-D)	Berger–Parker Index	Dominant Species	Brillouin Index	Mean Number of Helminth Species in Infracommunity Min–Max
Eurasian blue tit *Cyanistes careuleus* (67)	44.8 */30	13.7	1–157	6.0 *	5 T– 2, C–1, N–2	0.4	0.8	*Leucochloridium paradoxum*	0.02	1.0 1–2
Great tit *Parus major* (166)	62.7 */104	19.4	1–222	11.4 *	8 T–2, C–2, N–4	0.3	0.9	*Leucochloridium paradoxum*	0.03	1.1 1–3

n—number of infected birds, T—Trematoda; C—Cestoda; N—Nematoda; *—statistically significant difference between parameter of infection of Eurasian blue tit and Great tit, *p* ≤ 0.05.

**Table 3 animals-13-00421-t003:** Occurrence of helminths in juvenile and adult tits and in female and male tits. Numbers of examined juveniles and adults and females and males are given in brackets.

		Juveniles				Adults			Females			Males	
		Prevalence [%]/n	Mean Intensity	Mean Abundance	Prevalence [%]/n	Mean Intensity	Mean Abundance	Prevalence [%]/n	Mean Intensity	Mean Abundance	Prevalence [%]/n	Mean Intensity	Mean Abundance
*Cyanistes caeruleus*		(16)			(14)			(17)			(13)		
*L. paradoxum*		6.25/1	5	0.3	7.1/1	92	6.6	11.8/2	48.5	5.7	0.0	-	-
	*M. parinum*	0.0	-	-	0.0/0	-	-	0.0/0	0.0/0	0.0	0.0/0	-	-
	Cestoda gen. sp.	0.0	-	-	7.1/1	1	0.1	5.9	1	0.2	0.0/0	-	-
	Total Cestoda	0.0/0			7.1/1	1	0.1	5.9/1	1	0.2	0.0/0	-	-
	C. *pavlovskyi*	50.0/8	1.8	0.9	35.7/5	1.8	0.6	41.2/7	2.1	0.9	46.2/6	1.3	0.6
	Nematoda gen. sp.	0.0	-	-	7.1/1	4	0.3	5.9/1	4	0.2	0.0	-	-
	Total Nematoda	50.0/8	1.8	0.9	35.7/5	2.6	0.9	41.2/7	2.7	1.1	46.2	1.3	0.6
	Total helminths	50.0/8	2.4	1.2	42.9(6)	17.7	7.6	47.1/8	14.6	6.9	46.2/6	1.3	0.6
*Parus major*		(73)			(22)			(58)			(43)		
	*L. paradoxum*	12.3/9	62.1	7.7	9.1/2	32.5	2.9	19.0/11	63.5	12.1	11.6/5	30.2	3.5
	*U. macrostomus*	2.7/2	30.5	0.8	4.5/1	10	0.4	1.7/1	10	0.2	7.0/3	20.7	1.4
	Digenea gen sp.	1.4/1	1.0	0.0	0.0/0	-	-	1.7/1	1	0.0	0.0/0	-	-
	Total Digenea	16.4/12	51.8	8.5	13.6/3	25	3.4	22.4/13	54.6	12.2	16.3/7	30.4	5.0
	*A. globata*	0.0	-	-	4.5/1	2	0.1	1.7/1	2	0.0	0.0	-	-
	*M. parinum*	0.0	-	-	0.0	-	-	0.0	-	-	2.3/1	5	0.1
	Cestoda gen. sp.	2.7/2	-	-	27.3/6	-	-	12.1/7	-	-	4.7/3	-	-
	Total Cestoda	2.7/2 *	-	-	31.8/7 *	-	-	12.1/7	-	-	7.0/3	-	-
	*C. pavlovskyi*	52.1/38	2.8	1.4	36.4/8	3.6	1.2	44.8/26	2.6	1.1	51.2/22	3.1	1.6
	*D. obtusa*	0.0	-	-	4.5/1	-	-	1.7/1	1	0.0	0.0/0	-	-
	Nematoda gen sp.	1.4/1	1.0	0.0	0.0/0	-	-	0.0/0	-	-	0.0/0	-	-
	Total Nematoda	53.4/39	2.7	1.5	36.4/8	4.9	1.8	48.3/28	3.0	1.4	51.2/22	3.1	1.6
	Total helminths	60.3/44	16.5	10.1	68.2/15	3.6	1.9	70.7/41	20.8	13.9	55.8/26	11.3	6.6

n, number of infected birds, * statistically significant difference between infection of juveniles and adults, *p* ≤ 0.05.

**Table 4 animals-13-00421-t004:** Helminths recorded in Eurasian blue tit *Cyanistes caeruleus* and in Great tit *Parus major*. A slash (/) between species names indicates synonyms.

*Cyanistes caeruleus*	*Parus major*
DIGENEA	
*Leucochloridium paradoxum* Carus 1835 [36,48,49,50] and this study *Lutztrema attenuatum/ Brachylecithum attenuatum* (Dujardin, 1845) [48,51,52,53] *Plagiorchis maculosus* (Rudolphi, 1802) [48,49] and this study *Plagiorchis elegans* (Rudolphi, 1802) [48,51,52,53] *Prosthogonimus ovatus* (Rudolphi, 1803) [48,51,52,53,54,55]	
*Hypoderaeum* sp. Duetz, 1909 [56]	*Collyriclum faba* (Bremser, 1831) [51,52,53] *Cortrema magnicaudata/ Renicola magnicaudata* (Bykhovskaya-Pavlovskaya, 1950) [51,52,53] *Leucochloridium actitis /L. perturbatum* Pojmańska, 1969 [50] *Lyperosomum petiolatum /Lyperosomum platynosoides* (Railliet, 1900) [51,52,53,54] *Leyogonimus postgonoporos/ Maycella postgonoporus* Neiland, 1951 [48] *Plagiorchis laricola* Skrjabin, 1924 [51,52,53] *Plagiorchis notabilis* Nicoll, 1909 [51,52,53] *Plagiorchis multiglandularis* Semenov, 1927 [48] *Urogonimus macrostomus /Leucochloridium macrostomum* (Rudolphi, 1802) [48,51,52,53] and this study
CESTODA	
*Anonchotaenia globata* (von Linstow, 1879) [25,36,53] and this study *Emberizotaenia reductorhyncha* (Spasskaya 1957) [53] *Monosertum parinum* (Dujardin, 1845) [26,57] and this study	
*Anonchotaenia magniuterina* (Rysavy, 1957) [25,53] *Monopylidium praecox* (Krabbe, 1879) [56]	*Orthoskrjabinia bobica* (Clerc, 1903) [56] *Passerilepis passeris* (Gmelin, 1970) [53] *Passerilepis parina* (Fuhrmann, 1907) [57] *Passerilepis spasskii* (Sudarikov, 1950) [53] *Variolepis farciminosa* (Goeze, 1782) [53]
NEMATODA	
*Capillaria tridens* (Dujardin, 1845) [46,58] and this study *Diplotriaena henryi* (Blanc, 1919) [46,58] and this study *Cardiofilaria pavlovskyi* (Strom, 1937) [46,58] and this study	
	*Dispharynx nasuta* (Rudolphi, 1819) [46,59] *Diplotriaena obtusa* (Sonin 1968) [59] and this study *Serratospiculum* spp. (Skrjabin, 1915) [60] *Physocephalus sexalatus* (Molin, 1860), larvae [53]

## Data Availability

The raw data are available from the authors. The sequences from the study were uploaded to National Center for Biotechnology Information, GenBank accession MK882500-MK882508.

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
