# Peer review of "Helminth Community Structure of Tits *Cyanistes caeruleus* and *Parus major* (Paridae) during Their Autumn Migration on the Southern Baltic Coast"

_animals, 2023, doi:10.3390/ani13030421_

Round 1
Reviewer 1 Report
The manuscript “Helminth community structure of tits Cyanistes caeruleus and Parus major (Paridae) during their autumn migration on the southern Baltic coast” presented well prepared taxonomic and faunistic data regard to common representatives of tits (Paridae) form north part of Poland. And, in my opinion, after several corrections it will be able to be published.
Main comment /question
Helminthological studies have a long and rich history in Poland, the more surprising is the poor knowledge of tits’ parasites in this area. In the presented work, the Authors obtained a new parasite-host relationship for mentioned species of passerine birds. The Authors consider the presence of these parasites (or most of them) to be explained by the migration of the tit populations. However a question arises, is it for sure? and does this apply to all taxa discussed? bearing in mind the lack of broader data from other regions of the country. For example, Okulewicz (1991, 1997) obtained in the south-west part of Poland three Nematoda species from four detected in this study, and additionally, except Cardiofilaria pavlovskyi these other three species were sporadically reported in the actually presented work. Of course, the prevalence of C. pavlovskyi (very high) is most likely related to migration. And why migration is a factor influencing infection of L. paradoxum (conclusion in lines 374-377)? Maybe there are some data (not from Poland) regarding the occurrence of Leucochloridum sp. which would support this hypothesis.
Other comments/shortcomings
1. Introduction, lines 66-71. General is it true, but whether the migration of tits translates into the human parasites mentioned here.
2. Table 1. (a) Mean abundance or Relative abundance? (b) Values of “Cestoda-total” do not correspond with data of particular species of tapeworms…
3. Table 3. There are mistakes in presented values in the case M. parinum and Nematoda gen sp. , e.g. M. parinum - there are data for males but these males were adults or juveniles (here is “0”).
4. Table 4. Please correct the generic name of Lyperosomum (not Lyperostomum).
5. I suggest changing the part of sentences: lines 229-230 and 239-241. “nematodes was noted only…” and “nematodes only in males”, probably will be better (only proposition): “in juvenile Eurasian blue tit were found only nematodes…” and “ but male were infected only by nematodes”.
6. Sentence 384-385 is not exactly true. According to Rzad et al. 2011 (Annals of Parasitology) L. paradoxum was also recorded in Parus major.
Author Response
We would like to thank for all the comments in relation to the manuscript. Based on the comments we have applied following changes to the manuscript:
General part:
We did not aim to indicate that parasites found in the tits are directly related to the migration of birds, but to characterise parasite communities in migrating populations of C. careuleus and P. major. Therefore, according to the Reviewer’s comment in the Discussion we have modified the title of the subchapter as follows “4.2 Presence of parasites in the geographical distribution of C. careuleus and M. major” and additional clarification was added to the manuscript “Our study characterised diversity and abundance of internal parasites in the tits during period of their autumn migration. However, the presence of the parasites can not be explanyed solemny by the fact of bird migration, since parasites may be present both at the nesting and wintering sites that largly overlaps. Presence of parasites in tits does not stems from the migration itself, but from their distribution in the habitat area of tits. However, due to commonly observed clumped density of parasites in hosts and ecosystems, birds migrating between those areas can be invaded by diverse species and number of pathogens. Therefore, populations of birds that do not migrate and inhabit overlaping nesting and wintering areas can have diferent abundance and species diversity of parasires than birts that migrate. At this timepoint, comparison of parasite populations for C. careuleus and M. major is hindered due to lack of detailed information on the parasites of tits during nesting and wintering periods. Among the parasithes indentified in our study, high incidece of C. pavlovskyi in presumably related to migration of birds. In the 1971-1988 this species of nematode was identified in M. major nesting in Poland with prevalece of 6% and range of intensity 1-15 (Okulewicz 1991). Regarding the number of parasites in hosts, expressed by mean intensity or mean abundance, the high values might result from the environmental and physiological stress, but not only from the superinfective processes (Nowak and May 1994).”
In relation to the comment on L. paradoxum in the Discussion we have added the following information:
“As indicated in abovementioned works, trematodes of L. paradoxum are commonly present in Europe, including Poland. Thus, identifications of L. paradoxum in migrating tits, both in the earlier studies and in our work, does not indicate that migration of birds is the factor that influence infection of L. paradoxum, but only show that parasites are present in the birds during their migration.”
Ad. 1. „Introduction, lines 66-71”. In the Introduction we have added the following sentence „Migration of tits and its parasites does not translate directly to the human parasites. However, mechanisms of parasite transmission by vectors, influenced by global environmental changes, are common and related to the concept of “One Health”, i.e. healthy animals-healthy environment-healthy humanity, thus those pathogens have to be monitored.”
Ad. 2. „Table 1”: „(a)”: Names were provided according to Bush et al. (1997): „Mean abundance” (listed 33 in References) „(b)” Values indicated in the rows “Cestoda – total” values relate to the parasites identified up to the species level and individuals with no taxonomic assignment (details in the Materials and Methods). The sentence was added to the title of the Table 1.
Ad. 3. „Table 3”: Values related to the M. parinum and Nematoda gen sp. are correct. M. parinum was identified in male of unknown age (juvenile or adult), thus assignment of this male was impossible to a specific age group. Single specimen of Nematoda gen. sp. of unknown sex was found in juvenile P. major, thus assignment to a specific sex group was impossible. Tits with determined age and sex might have been different individuals. We have added specific clarification to the Material and Methods to increase readability.
Ad. 4. „Table 4”: The genus name of Lyperosomum was corrected.
Ad. 5. „Lines 229-230 and 239-241”: Updated as suggested by the Reviewer.
Ad. 6. Updated as suggested by the Reviewer and information on identification of L. paradoxum in P. major was added.
Reviewer 2 Report
The MS is well-written and quite complete in terms of the data, analyses, and interpretation. The authors should make more clear that parasite abundance/density in individuals can be high as a consequence of environmental or physiological stress, and not only by superinfective processes. Regardless, the differences between species, age, and location are well documented here.
More detail should be provided on the molecular identification of parasite species besides the Genbank acquistition.
Author Response
We would like to thank for all the comments in relation to the manuscript. Based on the comments we have applied following changes to the manuscript:
Ad. 1. in the Discussion we have added the following information:
„ Regarding the number of parasites in hosts, expressed by mean intensity or mean abundance, the high values might result from the environmental and physiological stress, but not only from the superinfective processes (Nowak and May 1994).”
- Additional information on molecular analysis of the parasites were added to the manuscript.